# Vitamin D Supplementation and Adherence to World Cancer Research Fund (WCRF) Diet Recommendations for Colorectal Cancer Prevention: A Nested Prospective Cohort Study of a Phase II Randomized Trial

**DOI:** 10.3390/biomedicines11061766

**Published:** 2023-06-20

**Authors:** Davide Serrano, Federica Bellerba, Harriet Johansson, Debora Macis, Valentina Aristarco, Chiara A. Accornero, Aliana Guerrieri-Gonzaga, Cristina M. Trovato, Maria Giulia Zampino, Emanuela Omodeo Salè, Bernardo Bonanni, Sara Gandini, Patrizia Gnagnarella

**Affiliations:** 1Division of Cancer Prevention and Genetics, European Institute of Oncology (IEO) IRCCS, 20141 Milan, Italy; harriet.johansson@ieo.it (H.J.); debora.macis@ieo.it (D.M.); valentina.aristarco@ieo.it (V.A.); chiaraarianna.accornero@ieo.it (C.A.A.); aliana.guerrierigonzaga@ieo.it (A.G.-G.); bernardo.bonanni@ieo.it (B.B.); 2Department of Experimental Oncology, European Institute of Oncology (IEO) IRCCS, 20141 Milan, Italy; federica.bellerba@ieo.it (F.B.); sara.gandini@ieo.it (S.G.); 3Division of Endoscopy, European Institute of Oncology (IEO) IRCCS, 20141 Milan, Italy; cristina.trovato@ieo.it; 4Division of Medical Oncology Gastrointestinal and Neuroendocrine Tumors, European Institute of Oncology (IEO) IRCCS, 20141 Milan, Italy; maria.zampino@ieo.it; 5Division of Pharmacy, European Institute of Oncology (IEO) IRCCS, 20141 Milan, Italy; eomodeo@ieo.it; 6Division of Epidemiology and Biostatistics, European Institute of Oncology (IEO) IRCCS, 20141 Milan, Italy

**Keywords:** Vitamin D, Vitamin D receptor, Vitamin D binding protein, colorectal cancer, adipokine, diet

## Abstract

Vitamin D and a healthy diet, based on World Cancer Research Fund (WCRF) recommendations, are considered key elements for colorectal cancer (CRC) prevention. In a CRC case-control study, we observed that CRC cases were often significantly Vitamin D deficient while subjects following WCRF recommendations significantly decreased their risk of developing CRC. We conducted a randomized phase-II trial (EudraCT number-2015-000467-14) where 74 CRC patients showed differences in response to Vitamin D supplementation, 2000 IU in average per day, according to gender and microbiota. The aim of this nested study is to correlate Vitamin D (supplementation, serum level and receptor polymorphisms), circulating biomarkers, and events (polyp/adenoma, CRC relapse and other cancers) in concomitant to WCRF recommendation adherence. Vitamin D supplementation did not modulate circulating biomarkers or follow-up events. *FokI* and *TaqI VDR* were associated with 25-hydroxyvitamin D (25OHD) levels. Patients following the WCRF recommendations had significantly lower leptin, significantly lower IL-6 (only in females), and significantly lower risk of events (HR = 0.41, 95%CI: 0.18–0.92; *p* = 0.03; median follow-up 2.6 years). Interestingly, no WCRF adherents had significantly more events if they were in the placebo (*p* < 0.0001), whereas no influence of WCRF was observed in the Vitamin D arm. While one-year Vitamin D supplementation might be too short to show significant preventive activity, a healthy diet and lifestyle should be the first step for preventive programs.

## 1. Introduction

Colorectal cancer (CRC) is among the most common malignancies. The majority (60–65%) can be defined as sporadic cancer, and it has been suggested that inflammation plays a causative role in its pathogenesis together with mechanisms to escape immune surveillance [1,2]. Risk factors are strongly related to unhealthy behaviors. An unhealthy dietary pattern, a sedentary lifestyle, and obesity are well-established CRC risk factors. More than specific nutrients or single food, the dietary pattern should be considered. A recent meta-analysis showed a relative risk (RR) for CRC of 1.25 (95% CI 1.11–1.40) for Western diets compared to healthy dietary patterns, RR 0.81 (95% CI 0.73–0.91) [3].

Healthy dietary patterns include mostly plant-based foods such as vegetables, fruits, whole grains, nuts, legumes, and moderate consumption of animal products (poultry, fish, and seafood), as opposed to high intakes of red and processed meat, sugar-sweetened beverages, and refined grains (overall high glycemic index food) [4]. According to the World Cancer Research Fund (WCRF), the established risk factors for CRC are alcohol consumption, body fatness, and processed and red meat consumption; in contrast, physical activity and foods containing fiber and calcium are protective factors [5]. Moreover, the consumption of foods containing or fortified with Vitamin D can be a protective factor for CRC. According to two recent dose-response meta-analyses, foods containing Vitamin D and Vitamin D supplementation showed a significant decreased risk (RR 0.95; 95% CI 0.93–0.98) and (RR 0.93; 95% CI 0.88–0.98), respectively. No effect was found for plasma and serum Vitamin D after evaluating 12 studies. The dose-response meta-analysis showed a borderline effect per 30 nanomoles/L (RR 0.92; 95% CI 0.85–1.00), even stratifying by sex, geographical location, and cancer site (https://www.wcrf.org/wp-content/uploads/2021/02/Colorectal-cancer-report.pdf). 25OHD levels have been related to body fat, a higher body mass index (BMI) was significantly associated with lower serum 25OHD levels [6], and a low vitamin D (25-hydroxy vitamin D, or 25OHD) level has been associated with cancer risk and other diseases in several observational studies [7,8]. Nevertheless, its causality for pathological processes remains uncertain, especially the role of Vitamin D supplementation. In randomized controlled trials (RCT) with Vitamin D supplementation versus placebo, the benefit of vitamin D was significant only for cancer survival and mortality rather than incidence [9]. Interestingly, the secondary analyses of the VITAL study [10] found a significantly reduced risk of all-cancer incidence for those with a BMI < 25 kg/m^2^. Furthermore, it is important to remember that VITAL and most vitamin D RCTs are designed using guidelines for pharmaceutical drugs rather than nutrients. The vitamin D dose used in the RTCs is probably too low to have an impact on cancer prognosis, and the inclusion criteria should be more precisely defined to identify the patients who may really benefit from vitamin D supplementation [11,12,13].

We also showed that Vitamin D supplementation could be considered an intervention (tertiary prevention) to improve survival in cancer patients, but further investigations are warranted [14]. Interestingly, a secondary analysis of a randomized clinical trial with 25,871 patients found that supplementation with vitamin D3 reduced the risk of incidence of advanced (metastatic or fatal) cancer, in particular in subjects with normal weight, but no reduction was observed among overweight or obese individuals. These results suggest a potential interaction between Vitamin D and BMI [15].

Vitamin D activity is mediated by the vitamin D receptor (VDR), and its polymorphisms may impair the target cells’ response to the hormone. Indeed, VDR is expressed in many tissues supporting the role of Vitamin D beyond bone metabolisms [16]. VDR polymorphisms have been described as being correlated with cancer risk, and for CRC, the BsmI polymorphism has consistently been reported to be associated with a reduced CRC risk [17,18]. Moreover, other genetic polymorphisms of enzymes involved in the Vitamin D metabolism, such as the vitamin D binding protein (VDBP), may also affect the bioavailability of 25OHD [19,20]. Furthermore, Vitamin D has been shown to exert anti-inflammatory activity and to be inversely associated with serum levels of CRP and Interleukin-6 (IL-6) [21]. An inverse relationship between Vitamin D and IL6 was also evident in our recent CRC case-control study [22]. Several studies have reported a possible interaction between Vitamin D and gut microbiota, strongly correlated with obesity and inflammation. A possible inter-player between the two is the immune system [23].

We first conducted a case-control study with CRC patients at the time of cancer diagnosis [22]. This study showed that a beneficial microbiota ratio (*Bifidobacteria/Escherichia* genera ratio) attenuates CRC risk due to an unhealthy diet. Subsequently, a phase II clinical trial was developed to investigate the microbiome changes with Vitamin D supplementation in CRC patients [24]. Participants were randomized to vitamin D 2000 UI per day or placebo for one year. The trial showed that Vitamin D supplementation can shape gut microbiota and that microbiota mediates the effect of supplementation on final 25(OH)D levels. We observed gender differences within Vitamin D metabolism, underlining that sex can be a key variable in studies where the role of Vitamin D and/or microbiota is investigated. Moreover, we found a significant association of the *FokI* variant with CRC (*p* = 0.03).

Here we present the results of a prospective cohort study, nested within the randomized trial described above. We analyzed circulating biomarkers and events (polyps/adenoma, CRC relapse and other cancers) in relation to adherence to WCRF recommendations and Vitamin D supplementation in CRC patients. Furthermore, we investigated the association of single nucleotide polymorphisms (SNPs) involved in vitamin D bioactivity and baseline circulating levels of 25(OH)D and vitamin D binding protein (VDBP).

## 2. Materials and Methods

The main endpoint of the present trial was recently published [24]. Briefly, the study was conducted from 2016 to 2019. Participants had a CRC diagnosis (stage I–III) and were treated accordingly. After completion of their standard treatment, they were randomized to Vitamin D 2000 IU per day (7 drops) versus placebo (7 drops) in a 1:1 ratio for 12 months. The rationale for the choice of this dosage was that the safe Recommended Daily Allowance is 2000 IU [25,26,27]. Furthermore, it has been calculated that with 2000 IU daily, only 10–15% of persons remain with a concentration < 30 ng/mL [27,28] and the results from a meta-analysis on 25(OH)D serum levels showed that 30 ng/mL is associated with a significantly lower risk of CRC [8]. The study was approved by the Institutional Review Board (the European Institute of Oncology Ethical Committee IEO-223, EudraCT number 2015-000467-14), and all subjects gave their written informed consent.

Seventy-four participants were included in the study. The flow diagram and baseline characteristics details of the study population were described in the previous publication [24]. Briefly, after confirmation of eligibility criteria, the participant was randomized to Vitamin D supplementation vs. placebo and stratified by chemotherapy (yes vs. no). Eleven patients received neoadjuvant treatment with pelvic irradiation (total of 50 Gy) and concomitant chemotherapy (9 patients received capecitabine in monotherapy and 2 patients fluoropyrimidine with oxaliplatin). Adjuvant treatment was received by 38 participants (18 received capecitabine in monotherapy, and 20 received a fluoropyrimidine and oxaliplatin regime). After treatment completion and subsequent minimum 6 months wash-out, patients were invited to participate in the study. At the baseline visit, medical history, concomitant medications, food consumption, clinical examination, and anthropometric measurements were acquired. Fasting blood and fecal samples were collected. Follow-up was completed either through a clinical visit or a phone-call contact. After the year of study intervention, participants continued their annual oncology visits. Follow-up data were collected by their chart or by phone call contact for those who continued their follow-up visits in a different hospital.

### 2.1. Circulating Biomarkers

At baseline, serum 25(OH)D concentrations were determined by a commercially available chemiluminescent immunoassay designed for the IDS-iSYS automated instrument (Immunodiagnostic Systems, Pantec S.r.l., Turin, Italy). This method recognizes both metabolites of vitamin D (D2–D3), and correlates well with the isotope-dilution liquid chromatography-tandem mass spectrometry (ID–LC-MS/MS) method [29], without any statistically significant bias. Vitamin D Binding Protein (VDBP) was determined by ELISA (R&D Systems Europe, Ltd., Abingdon, UK), while IL-6, IL-10, leptin, and adiponectin were determined using an automated immunoassay platform called ELLA (ProteinSimple, Bio-techne, Minneapolis, MN, USA). All assay runs included pooled serum control samples to monitor the coefficient of inter-assay variability. This variability never exceeded 11%. To reduce the effect of technical variability, baseline and follow-up samples from each subject were processed next to each other.

### 2.2. Genotyping Biomarkers

Genomic DNA was extracted from whole blood samples with a QIAamp DNA blood kit (Qiagen, Valencia, CA, USA) according to the manufacturer’s instructions by the automated platform “QIAcube” (Qiagen, Valencia, CA, USA) and quantified using a NanoDrop spectrophotometer (Thermo Scientific, Wilmington, DE, USA). DNA samples were genotyped for a comprehensive set of single nucleotide polymorphisms (SNPs) by the use of TaqMan SNP genotyping assays run on an ABI PRISM 7500 FAST Real-Time PCR System (Thermo Fisher Scientific Wilmington, DE, USA).

We analyzed *BsmI* (rs1544410), *TaqI* (rs731236), *FokI* (rs2228570), *ApaI* (rs7975232), and *CDX2* (rs11568820) in the VDR gene; *CYP24A1*-rs6013897, *CYP27B1*-rs10877012, *CYP2R1*-rs10741657, genes involved in Vitamin D metabolism; and rs2282679, and rs4588 in the GC gene coding for the VDBP. Briefly, nearly 10 ng of DNA in 2 μL was added to an 8-μL reaction well, together with 10 μL of reaction mix containing forward and reverse primers and two allele-specific fluorescent-labeled probes (one wild-type and one variant allele-specific).

### 2.3. Food Consumption

Food consumption was evaluated using a short questionnaire adapted from a validated questionnaire [30]. The questionnaire evaluates the main food groups commonly consumed by the Italian population. Moreover, a specific question was adapted to better discriminate food with a potential source of vitamin D. The frequency of the consumption of food is grouped into five levels, from “never or seldom” to “high frequency”, on a daily or weekly basis to assess average consumption. For each item, the standard portion size was indicated to obtain as accurate an answer as possible. To identify a protective pattern according to the WCRF’s recommendations (WCRF 2018 https://www.wcrf.org/diet-activity-and-cancer/cancer-prevention-recommendations), we built a score taking into account body weight (BMI), the level of physical activity, and dietary habits. The score inversely associated with CRC [22] was characterized by a high level of physical activity, a normal range of BMI, and a healthy pattern of high consumption of fruit and vegetables, or low consumption of meat or sweets, cakes, and pastries.

### 2.4. Statistical Analysis

A patient was considered to be adherent to the WCRF recommendations when he/she was in the normal range of baseline BMI (BMI < 25), practiced a high level of physical activity, and had a healthy diet (high consumption of fruit and vegetables, or low consumption of meat or sweets, cakes, and pastries). Differences in baseline serum biomarkers by WCRF adherence and gender and by VDR and VDBP variants were assessed through a non-parametric Wilcoxon rank test.

For the Event-Free Rate (EFR) analysis, the time-to-event was calculated as the difference between the date of the first event and the randomization date for those patients in whom at least one event occurred (colorectal adenoma, cancer relapse, or death), and as the difference between the date of last visit and the randomization date in those in whom no event of progression occurred. Comparisons in EFR by adherence to the WCRF’s recommendations were carried out using the Kaplan–Meier estimator and tested with the log-rank test. Multivariable Cox proportional-hazards models were employed to estimate the risk of the event in terms of hazard ratios (HRs); 95% confidence intervals (CI) were also provided.

## 3. Results

Seventy-four patients were enrolled in the ColoViD trial, 36 in the placebo group and 38 in the Vitamin D group, respectively. Their main characteristics were: average age was 62 years old, 53% were female, 60% of original cancers were stage II–III, and 55% were G2; 37 patients underwent chemotherapy (either as neoadjuvant or adjuvant treatment, for more details see the material section). Overall, the two arms were well balanced. Other descriptive features of the cohort can be found in the previous publication [24].

The Vitamin D supplementation did not significantly modulate leptin and adiponectin, nor the other analyzed circulating biomarkers (see Appendix A).

Based on the evaluation of the dietary pattern and lifestyle characteristics at baseline, we could categorize the population in two groups: “adherent” to WCRF recommendations versus “non-adherent”. Leptin and BMI were found to be significantly different between patients who adhered or did not adhere to the WCRF recommendations. Table 1 shows lower leptin (*p* = 0.001 and *p* = 0.003 for females and males, respectively) and lower BMI (*p* = 0.0003 and *p* = 0.0096 for females and males). Furthermore, we found borderline significantly higher 25(OH)D levels (*p* = 0.059 in males) in patients following WCRF indications. IL-6 was significantly higher in women who did not adhere to the WCRF recommendations (*p* = 0.046), while no significant differences were observed in men (*p* = 0.242).

The circulating levels of VDBP were highly correlated with the SNPs of the *GC* gene and several SNPs of the VDR. Specifically, *FokI* and *TaqI*, were associated with 25(OH)D levels (Table 2 and Table 3). As we did not observe any associations of the enzymes involved in the 25(OH)D metabolism, we did not include this information in this table.

The multivariable Cox proportional hazard model also confirmed a lower risk (HR = 0.41; 95% CI 0.18–0.92 *p* = 0.03, Table 4), adjusting for trial arms.

During a median follow-up of 2.6 years, we found 31 events: 6 CRC relapses (8.1%), 21 adenoma/polyps (28.4%), and 4 other cancers (5.4%). Adherence to WCRF was found to be associated with a significantly lower risk of any events (Figure 1 Kaplan–Meier curve for EFR; Log-rank *p* = 0.01).

Interestingly, the analysis per study arms showed even greater reduced risk in the WCRF adherent participants within the placebo group. On the other hand, in the treatment arm no differences were seen in respect to diet adherence (Figure 2). These results suggest a possible interaction between Vitamin D and WRCF adherence: a lower risk of events is observed in the WCRF adherent group not taking vitamin D or in the vitamin D arm, independently of WCRF adherence. 

## 4. Discussion

This exploratory study is nested within a phase II trial assessing the effect of Vitamin D supplementation on microbiome change [18]. Our data showed that CRC patients who adhere to WCRF recommendations had a significantly lower risk of events such as new polyps/adenoma, CRC relapse and other cancers compared to those with poor dietary habits. Interestingly, patients who follow WCRF recommendations have a significantly lower level of leptin; male participants have a greater level of 25OHD, and females have a lower level of IL-6. Adherence to such guidelines seems to produce a healthier profile of inflammation biomarkers and hormonal response, driven mainly by compliance with recommendations on body fatness, physical activity, and energy-dense foods and drinks as reported by other research [19]. 

The effect on colorectal events suggests a possible interaction between Vitamin D and diet: a lower risk of events is observed both in the WCRF adherent group not taking vitamin D and in the vitamin D arm, independently of WCRF adherence. Since the benefit due to the WCRF adherence was no longer evident in the Vitamin D arm, these data underline the role of both vitamin D supplementation and lifestyle indications for cancer prevention. It has been established that fat tissue that produces and secretes hormones, including leptin and adiponectin, is strongly involved in cancer risk [31,32]. Leptin is important in energy balance and appetite control and positively correlated with adipose tissues and nutritional status, and more recently has been extensively studied as a potential mediator of obesity-related cancer [33]. Moreover, leptin plays a key role in inflammation due to a large variety of metabolic effects, and increases both fatty acid oxidation [34] and glucose uptake [35]. However, chronic inflammation could downregulate immune system functions, producing homeostatic changes and affecting lipid metabolism [36].

Inflammation certainly has cancer-promoting effects. We found that IL-6 was significantly higher in women who did not adhere to the WCRF recommendations. IL-6 seems to be a valid marker of colorectal inflammation as reported by Kakourou et al. [37]. Diet can be pro-inflammatory, and it has been shown in a prospective cohort study that higher inflammatory dietary patterns in association with supplements increase the risk of adenoma recurrence and CRC incidence. These data would suggest that subjects with a history of adenoma should follow an anti-inflammatory diet and avoid unmotivated supplements [38]. Vitamin D supplementation has an effect on circulating cytokine and their modulation can be different based on the individual condition, being more pronounced in patients with inflammatory disorders compared to healthy subjects [39,40]. Furthermore, the trial by Fassio et al. did not show significant differences among the different Vitamin D dosages. To note, in Fassio’s study recruited subjects had vitamin D levels below 20 ng/mL (deficiency), while in our study the participants were below 30 ng/mL.

The impact of the Vitamin D treatment did not show any effects on the analyzed circulating biomarkers; however, we found that men with higher WCRF adherence have borderline significantly greater 25(OH)D serum levels. Several SNPs of the VDR gene have been associated with cancer, including CRC [41]. Different VDR domains are involved in several functions, including DNA binding, receptor dimerization, gene transactivation, and cofactor activation. The findings from our study regarding its association with 25(OH)D may play a role in cancer development. Low 25(OHD)D levels, its metabolisms, and VDR polymorphisms play a central role in Inflammatory Bowel Disease (IBD) pathogenesis [42], and IBD is a CRC risk factor. However, the effect of vitamin D supplementation to induce a therapeutic impact has still many open questions, including timing compared to the pathological project, duration, and optimal dosage [29]. Previous studies reported that plasma 25(OH)D concentrations seem to be affected by the food pattern followed by study participants. Crowe found a lower level of plasma 25(OH)D in vegetarians and vegans than in meat and fish eaters from the European Prospective Investigation into Cancer and Nutrition (EPIC)–Oxford cohort [43].

This is an exploratory study describing the results of secondary endpoints of a study presented previously [24]. Its main limitation is the small sample size, which does not allow enough statistical power for our results, especially for gender subgroup analysis. Another issue is the choice of dose of 2000 IU/day that it may be too low [44]. However, all participants receiving the active treatment reached the sufficient level of vitamin D (>30 ng/mL). Moreover, one-year Vitamin D supplementation might be too short to have a clinical impact on the events since it takes time to reach a sufficient plasma level and make its beneficial effects explicit [44,45]. The dietary pattern of the WCRF adherence group was a personal choice that could reasonably be adopted for a longer time compared to the Vitamin D supplementation, leading to a clinical effect. This study is exploratory and did not evaluate a specific intervention on lifestyle, except for some general indications at baseline. For these reasons, the results of this study need to be confirmed in a randomized trial on lifestyle interventions, considering also gender differences.

The increasing cancer incidence and the number of long-term survivors underscore the need to promote and improve prevention projects. Cancer prevention programs have to meet the challenge of overcoming the lack of validated biomarkers as the readout of the intervention’s efficacy. Adipokines could be promising biomarkers, and adiponectin in particular showed an inverse correlation with breast cancer risk and relapse, specifically in a cohort study of premenopausal women [46] and in a meta-analysis [47]. Adiponectin was also found to be inversely associated with CRC risk [48] and relapse [49]. On the other hand, leptin was found to be associated with a higher risk of CRC, specifically in a meta-analysis of prospective studies [50].

## 5. Conclusions

This study supports the implementation of lifestyle intervention programs as approximately 45% of CRCs in Western countries can be related to modifiable lifestyle risk factors [51]. To expand the knowledge in cancer prevention, lifestyle educational and gender-oriented programs are needed in cancer and screening centers and should be offered by family doctors. Moreover, vitamin D supplementation confirms its potential as a preventive agent.

## Figures and Tables

**Figure 1 biomedicines-11-01766-f001:**
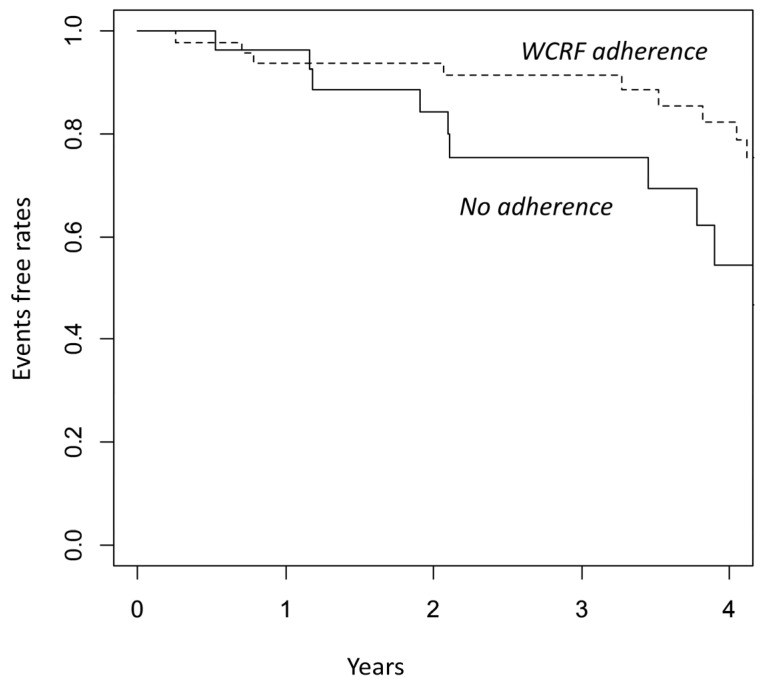
Kaplan–Meier for events according to dietary pattern: adherence to WCRF versus non-adherence (*p*-value = 0.013, Log-rank test).

**Figure 2 biomedicines-11-01766-f002:**
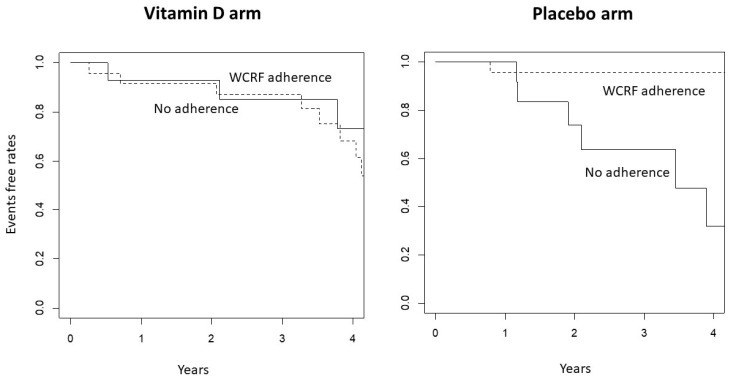
Kaplan–Meier for events according to dietary pattern: adherence to WCRF versus non-adherence by treatment arm (*p*-value = 0.97, in the Vitamin D arm and *p* < 0.0001 in the placebo arm Log-rank test).

**Table 1 biomedicines-11-01766-t001:** Baseline serum biomarkers levels and BMI by dietary pattern and gender.

			No WCRF Adherents				WCRF Adherents			
Gender	Variable	n.	Median	1st Q	3rd Q	n	Median	1st Q	3rd Q	*p*-Values
Females	BMI	12	28.87	26.64	30.62	22	23.32	20.70	24.24	**<0.001**
	25(OH)D ng/mL		20.0	11.2	29.5		22.1	18.7	25.1	0.98
	VDBP µg/mL		278	264	316		305	282	361	0.20
	Adiponectin µg/mL		12.39	11.21	14.11		11.97	9.83	20.16	1.00
	Leptin ng/mL		52.29	32.94	68.92		19.03	7.77	31.51	**0.007**
	IL-10 pg/mL		2.18	1.99	2.70		2.12	1.90	3.21	0.97
	IL-6 pg/mL		4.44	1.99	5.94		2.01	1.38	3.45	**0.046**
Males	BMI	15	29.01	26.59	32.51	25	25.88	24.22	27.36	**0.011**
	25(OH)D ng/mL		16.1	13.5	24.3		24.1	18.4	28.2	0.059
	VDBP µg/mL		268	248	308		302	258	327	0.21
	Adiponectin µg/mL		5.50	4.39	8.50		8.52	5.55	12.06	0.08
	Leptin ng/mL		21.17	11.08	35.81		6.92	5.04	11.27	**0.003**
	IL-10 pg/mL		2.41	1.69	2.89		2.33	1.86	2.65	0.88
	IL-6 pg/mL		2.52	2.12	3.51		2.04	1.6	4.31	0.24

In bold significant or borderline significant *p*-values.

**Table 2 biomedicines-11-01766-t002:** Median values and interquartile range of baseline serum levels of 25(OH)D (ng/mL) by variants of VDBP and VDR genes.

SNPs	Variants	N.	Median of Serum 25(OH)D (ng/mL)	1st Quartile	3rd Quartile	*p*-Values
GC rs2282679	GG	6	21	15	24	0.73
	TG	26	22	18	27	
	TT	42	23	14	26	
GC rs4588	GG	42	23	14	26	0.66
	GT	28	23	19	27	
	TT	4	18	15	23	
VDR Fokl	AA	9	22	16	24	**0.01**
	GA	32	24	22	28	
	GG	33	19	14	24	
VDR Taql	AA	25	20	14	24	**0.04**
	AG	36	21	16	25	
	GG	13	26	24	28	
VDR CDX2	CC	38	21	14	25	0.43
	CT	34	22	19	26	
	TT	2	24	22	27	
VDR BsmI	CC	22	20	14	25	0.12
	CT	38	22	16	25	
	TT	14	25.8	20	27.6	
VDR ApaI	AA	26	24.	19	28	0.23
	AC	34	20	14	25	
	CC	14	22	14	26	

*p*-values from Kruskal–Wallis Tests for the association between serum levels of 25(OH)D with VDR and VDBP genes variants. In bold significant or borderline significant *p*-values.

**Table 3 biomedicines-11-01766-t003:** Median values and interquartile range of baseline serum levels of VDBP (µg/mL) by variants of VDBP and VDR genes.

SNPs	Variants	N.	Median of Serum VDBP (µg/mL)	1st Quartile	3rd Quartile	*p*-Values
GC rs2282679	GG	6	217	197	234	**0.0004**
	TG	26	287	263	325	
	TT	42	303	270	361	
GC rs4588	GG	42	303	270	361	**0.002**
	GT	28	280	255	320	
	TT	4	201	197	219	
VDR Fokl	AA	9	302	267	310	0.92
	GA	32	285	268	333	
	GG	33	292	263	325	
VDR Taql	AA	25	298	258	319	0.72
	AG	36	289	266	328	
	GG	13	284	268	333	
VDR CDX2	CC	38	280	258	31	**0.06**
	CT	34	305	266	375	
	TT	2	338	334	343	
VDR BsmI	CC	22	290	241	310	0.44
	CT	38	289	267	326	
	TT	14	294	268	334	
VDR ApaI	AA	26	311	268	359	0.17
	AC	34	284	267	319	
	CC	14	275	230	302	

*p*-values from Kruskal–Wallis Tests for Vitamin D Binding Protein (VDBP) with VDR and VDBP genes variants. In bold, significant or borderline significant *p*-values.

**Table 4 biomedicines-11-01766-t004:** Hazard Ratio and 95% Confidence Interval for events from multivariable Cox proportional hazard model *.

	HR (95% CI)	*p*-Values
Vitamin D vs. placebo arm	1.20 (0.56; 2.57)	0.64
WCRF adherence (Yes vs. No)	0.41 (0.18; 0.92)	0.03

* EFR analysis includes: 6 relapses (8.1%), 21 adenoma/polyps (28.4%) and 4 other cancer sites (5.4%).

## Data Availability

Data on serum biomarkers and dietary information are available from the corresponding author on reasonable request.

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
