# Peer review of "Vitamin D Supplementation and Adherence to World Cancer Research Fund (WCRF) Diet Recommendations for Colorectal Cancer Prevention: A Nested Prospective Cohort Study of a Phase II Randomized Trial"

_biomedicines, 2023, doi:10.3390/biomedicines11061766_

Round 1
Reviewer 1 Report
Meat has vitamin D as 25(OH)D
Plasma concentrations of 25-hydroxyvitamin D in meat eaters, fish eaters, vegetarians and vegans: results from the EPIC-Oxford study.
Public Health Nutr. 2011 Feb;14(2):340-6. doi: 10.1017/S1368980010002454.
In randomized controlled trials (RCT) with VD supplementation versus placebo,
the benefit of vitamin D was significant only for cancer survival and mortality rather
than incidence [7]
Comment: The secondary analyses of the VITAL study [Manson, 2019] found a significantly reduced risk of all-cancer incidence for those with BMI <25 kg/m2. The problem with VITAL and most vitamin D RCTs is that they were designed using the guidelines for pharmaceutical drugs rather than nutrients, and too little vitamin D was given. See:
Guidelines for optimizing design and analysis of clinical studies of nutrient effects.
Nutr Rev. 2014 Jan;72(1):48-54. doi: 10.1111/nure.12090. Epub 2013 Dec 13.
Critical Appraisal of Large Vitamin D Randomized Controlled Trials.
Nutrients. 2022 Jan 12;14(2):303. doi: 10.3390/nu14020303.
The rationale for the choice of this dosage was that the safe Recommended Daily Allowance
is 2000 IU [35–37].
Comment: That dose is too low and the references from Vieth seem to suggest higher doses.
See:
Daily oral dosing of vitamin D3 using 5000 TO 50,000 international units a day in long-term hospitalized patients: Insights from a seven year experience.
J Steroid Biochem Mol Biol. 2019 May;189:228-239. doi: 10.1016/j.jsbmb.2018.12.010.
Intratrial Exposure to Vitamin D and New-Onset Diabetes Among Adults With Prediabetes: A Secondary Analysis From the Vitamin D and Type 2 Diabetes (D2d) Study. Diabetes Care 2020;43:2916-2922.
Diabetes Care. 2021 May;44(5):e106. doi: 10.2337/dci21-0006.
The Association between Serum 25(OH)D Status and Blood Pressure in Participants of a Community-Based Program Taking Vitamin D Supplements.
Nutrients. 2017 Nov 14;9(11):1244. doi: 10.3390/nu9111244.
Maternal 25(OH)D concentrations ≥40 ng/mL associated with 60% lower preterm birth risk among general obstetrical patients at an urban medical center.
PLoS One. 2017 Jul 24;12(7):e0180483. doi: 10.1371/journal.pone.0180483.
Breast cancer risk markedly lower with serum 25-hydroxyvitamin D concentrations ≥60 vs <20 ng/ml (150 vs 50 nmol/L): Pooled analysis of two randomized trials and a prospective cohort.
PLoS One. 2018 Jun 15;13(6):e0199265. doi: 10.1371/journal.pone.0199265.
The VD supplementation did not significantly modulate leptin and adiponectin nor
the other analyzed circulating biomarkers (see supplementary table1).
Comment: Vitamin D supplementation been shown to increase IL-6 and IL-10 in subjects with inflammatory disease.
The Effect of Weekly 50,000 IU Vitamin D3 Supplements on the Serum Levels of Selected Cytokines Involved in Cytokine Storm: A Randomized Clinical Trial in Adults with Vitamin D Deficiency.
Nutrients. 2023 Feb 27;15(5):1188. doi: 10.3390/nu15051188.
However, no effect on healthy subjects:
Effects on Serum Inflammatory Cytokines of Cholecalciferol Supplementation in Healthy Subjects with Vitamin D Deficiency.
Nutrients. 2022 Nov 14;14(22):4823. doi: 10.3390/nu14224823.
Cancer is not an inflammatory disease, so the finding re IL-6 and IL-10 is as expected.
Suggest not using “VD”. In English, it stands for venereal disease. Perhaps VitD?
by ELISA (R&D Systems).
Please provide city and country for all companies.
Under limitations of this study, low vitamin D dose should be mentioned and discussed.
Table 1. Please format the table so that the end letters to not spill over to the next line.
The header on every page uses 2021; it is now 2023.
Significant digits. The general rule is that no more non-zero digits should be given than are justified by the uncertainty of the value.
See "Too many digits: the presentation of numerical data"
https://www.ncbi.nlm.nih.gov/pmc/articles/PMC4483789/
If the uncertainty is greater than about 7%, only two non-zero digits are justified.
P values should be given to two decimal places unless the first two are 00 or the number lies between 0.045 and 0.054. If the first two are 00, then only one non-zero digit can be given.
Thus, none of the numbers in Tables 1 and 3 should have more than three digits.
Percentages should be given in whole numbers.
Please revise the P values as stated above.
Please review all numbers in abstract, text, tables, and figures and adjust accordingly.
Reviewer 2 Report
Vitamin D supplementation and adherence to World Cancer Research Fund (WCRF) diet recommendations for colorectal cancer prevention: a nestedprospective cohort study of a phase II randomized trial, presents results from a second sub-study within the trial. Here adherence to WCRF diet guidelines and vitamin D supplementation are evaluated for impact on adverse outcomes following treatment for CRC.
My main comment is to suggest more detail in Fig 1 or possibly addition of a second panel. I would like the vitamin D intervention to be included in the Figure. We would have the original KM plot, then each group, WCRF adherent and non-adherent broken down by VD intervention or placebo.
The English is passible. Just go through one more time to polish it.
Round 2
Reviewer 1 Report
Comment: The number of participants should be stated in the abstract, the length of the trial, the vitamin D dose given, and what definition of CRC event was used. VitD should be defined with the first time vitamin D is used, then used after that in place of vitamin D. Same rule for VitD in the text. There are currently 68 “vitamin D” and 38 “VitD” in the manuscript. Perhaps VitD could be omitted in favor of vitamin D?
The aim of this nested study, is to correlate VitD
Comment: is that vitamin D supplementation?
After completion of their standard treatment, they were randomized
to 25-OHD 2000 IU per day
Comment: No, it was vitamin D3 (cholecalciferol), not calcifediol.
During a median follow-up of 2.6 years, we found 31 events: 6 CRC relapses (8.11%),
21 adenoma/polyps (28.4%), and 4 other cancers (5.4%). Adherence to WCRF was found
to be associated with a significantly lower risk of any events (Figure 1 Kaplan-Meier curve
for EFR; Log-rank P=0.014).
Comment: There should be a table with the events for treatment and control giving the date and type of each event including what type of cancer if included in the analysis. Since it can take some time for serum 25(OH)D concentration to rise after starting supplementation (Heaney, 2003) and then some time to have an effect on cancer outcomes, if there were early events in the treatment arm, they may not be related to vitamin D supplementation (Lappe, 2007).
Minor point: percentages should be given in whole numbers and P = 0.01.
Human serum 25-hydroxycholecalciferol response to extended oral dosing with cholecalciferol.
Am J Clin Nutr. 2003 Jan;77(1):204-10. doi: 10.1093/ajcn/77.1.204.
Vitamin D and calcium supplementation reduces cancer risk: results of a randomized trial.
Am J Clin Nutr. 2007 Jun;85(6):1586-91. doi: 10.1093/ajcn/85.6.1586.
Figure 1. Kaplan-Meier for colorectal events according to dietary pattern: adherence to WCRF versus
non-adherence (P-value=0.013, Log-rank test).
Comment; P = 0.01
Later:
Since the sample size is small, as exploratory analysis
we considered a “colorectal event” as either a new adenoma or any other CRC relapse.
Question: Were the 4 other cancers considered in the analysis for Figures 1 and 2?
The two paragraphs do not agree.
To identify a protective pa?ern
according to WCRF recommendation (WCRF 2018 h?ps://www.wcrf.org/diet-activityand-
cancer),
Comment: That URL is not specific enough
Perhaps use https://www.wcrf.org/diet-activity-and-cancer/cancer-prevention-recommendations/
The main problem regarding the writing is the use of VitD as mentioned in the comments.
Reviewer 2 Report
The revised paper is acceptable for publication.
Author Response
No reply is needed
Round 3
Reviewer 1 Report
The manuscript has been improved.
A few more changes need to be made
Vitamin D serum levels. Several SNPs
Should be 25(OH)D serum levels
On the other hand, in the VitDVitamin D no differences
comment should be
On the other hand, in the Vitamin D supplementation or treatment arm, no differences
It appears that VitD is used in the abstract but vitamin D is used in the text. One or the other form should be used throughout. This and other errors indicates that the manuscript should be edited by a qualified medical journal manuscript editor.
